# Detection and Localization of Changes in Conditional Distributions

**Lizhen Nie**
Department of Statistics
The University of Chicago
lizhen@uchicago.edu

**Dan Nicolae**
Department of Statistics
The University of Chicago
nicolae@statistics.uchicago.edu

## Abstract

We study the change point problem that considers alterations in the conditional distribution of an inferential target on a set of covariates. This paired data scenario is in contrast to the standard setting where a sequentially observed variable is analyzed for potential changes in the marginal distribution. We propose new methodology for solving this problem, by starting from a simpler task that analyzes changes in conditional expectation, and generalizing the tools developed for that task to conditional distributions. Large sample properties of the proposed statistics are derived. In empirical studies, we illustrate the performance of the proposed method against baselines adapted from existing tools. Two real data applications are presented to demonstrate its potential.

## 1 Introduction

The canonical setting for change point (CP) analysis is concerned with the detection and localization of changes in distributions of an independent sequence of observations $\{x_t, t = 1, 2, \cdots, n\}$. In many applications, however, the observation at each index $t$ is a *pair* $(x_t, y_t)$ with $x_t$ a set of covariates and $y_t$ a variable of special interest for inference. This setting incorporates many important problems in a variety of scientific fields. For example, in finance, market data are often affected by monetary policies. Thus, to more accurately predict stock prices using this additional information, one may want to first understand whether and when the relationship between the target stock price and its associated covariates changes over time [37, 38, 28, 29, 49]. Another example comes from environmental analysis [6], where one goal is to learn the relationship between the levels of pollutants and the number of weekly hospital admissions for circulatory and respiratory diseases, which is subject to potential changes. Before considering more complex models that incorporate time-varying relationships, one needs to understand first whether the relationship indeed changes and if so, when.

Despite the existence of many applied scenarios where the relationship between a pair changes, change point methods specifically designed for paired data are quite scarce. As far as we know, existing methods assume univariate $y$'s and aim at changes in the conditional expectation of $y$ given $x$. Moreover, most papers [26, 24, 25, 39, 2, 4, 22, 43, 15, 40, 11] focus on the setting where the conditional expectation of $y$ depends linearly on $x$. The nonlinear and nonparametric cases, however, are relatively under-investigated. The existing methods mainly focus on two settings. One setting arises from challenges in econometric modelling, and they focus exclusively on the problem of testing for structural stability of the conditional mean function over time series [49, 19, 47, 46], while the (perhaps more) important problem of localizing the change point(s) is not investigated. Another setting assumes that the covariate $x_t$'s are fixed, univariate, and form an equi-spaced sequence lying in interval $[0, 1]$ [31, 32]. A detailed review of these related works is included in Section 2.

We are interested here in the case where both $x$ and $y$ can lie in a general space (not necessarily the real line), and we aim to identify changes in the whole conditional distribution, rather than only

the mean. This is of significance to many applications: for example, in finance, it is beneficial to understand changes in the volatility level of stock prices. In this paper we want to both identify and localize such changes. We will assume a random design setting. Contrary to the fixed design setting in [31, 32], we assume that the covariate $x_t$'s are randomly drawn from some unknown distribution. This assumption is realistic and meaningful. Consider the example where a sociologist wants to investigate whether the relationship between education and income has changed over years. A survey is distributed every year, which records both the covariate (education level) and the response (income). In this case, the covariates cannot be fixed (as one cannot determine who will take the survey).

In the random design setting, one could argue that, if we combine $x$ and $y$ and treat them as as *one* variable, a sufficiently powerful change point method that can handle *any* change in distribution should perform well. This approach is straightforward to implement; however, it targets changes in the joint distribution $p(x, y) = p(x)p(y \mid x)$, while we are only interested in changes in the conditional distribution $p(y \mid x)$. An intriguing question is whether a method especially designed for change points in this *conditional* distribution will have better performance than those for joint distributions. As the first work to study these general changes, we answer this question by: (i) developing a mathematical formulation of this new problem and establishing its connection to a simplified task; (ii) investigating the advantages and limitations of applying existing methods to this new problem; and (iii) proposing new methodology.

## 2 Related Work

This section reviews relevant CP literature on *paired* observations. Please refer to [34, 1] for reviews on the canonical (unpaired) setting. More related work is discussed in the Appendix Section B.

**Linear models.** Most publications on paired data assume that the conditional expectation of $y$ is a linear combination of $x$ [26, 24, 25, 39, 2, 4, 22, 43, 15, 40, 11], and attention has been focused on analyzing changes in the slope. Recently, generalizations of this classic linear model have been investigated. One strand of work focuses on analyzing changes in high dimensional linear regression models [30, 42, 50], while others include seasonality and/or correlated errors[43].

**Structural stability test for conditional expectation.** Several econometrics papers [49, 19, 47, 46] focus on testing the stability of the conditional mean function using nonparametric methods. Their task is simpler, as only testing is concerned, and moreover, only changes in the conditional expectation are considered. Direct generalizations of their methods to our problem are difficult, but they provide a good starting point.

**Nonparametric methods for fixed design.** [31, 32] consider univariate $x_t, y_t$'s, where $x_t$ forms an equi-spaced sequence, and they focus on changes in $\mathbb{E}[y_t \mid x_t]$. Furthermore, [32] assumes that direction of the change is known.We note that, although the fixed and random design settings seem similar, a method which works well for one might not even work for another. Intuitively, the fixed design setting can be formulated as finding break points in an otherwise continuous/smooth curve, which is different from our focus.

**Trend filtering.** The recently proposed trend filtering [27, 48, 51] also targets estimating conditional expectation in a fixed design. By solving a penalized least squares optimization problem, the estimated curve exhibit the structure of a piecewise polynomial function [48], the form of which is more restrictive than [31, 32].

**Bayesian approaches.** Bayesian methods are also used for CP problems. The most relevant are based on the Gaussian process change point analysis [44, 7, 23], which aims to detect change(s) in the mean and/or the covariance function of the Gaussian process regression. In order to derive a tractable posterior, a Gaussian prior is usually placed on the mean functions. The parametric formulations needed for Gaussian process CP approaches make them more restrictive than the methods proposed in this work.

## 3 Problem Statement

**Task I: Analyzing changes in conditional distributions.** The ultimate goal is to analyze changes in the conditional distribution for a paired sequence $\{(x_t, y_t), t = 1, 2, \cdots, n\}$ where $x_t \in \mathcal{X}$ and $y_t \in \mathcal{Y}$ with $(\mathcal{X}, d)$ a semi-metric space and $\mathcal{Y}$ a general space. We assume that the covariates $x_t \overset{\text{iid}}{\sim} F_X$, while the conditional distribution of $y_t$ given $x_t$ might go through an abrupt change.

Denote $\{F^0_{Y|X=x}, x \in \mathcal{X}\}$, $\{F^1_{Y|X=x}, x \in \mathcal{X}\}$ to be two sets of conditional distributions that differ on at least one $x \in \mathcal{X}$. Our task is to investigate:

- (Detection) Testing the null hypothesis

$$H_0 : \quad y_t \mid x_t \sim F^0_{Y|X=x_t}, \quad t = 1, 2, \cdots, n$$

against the alternative

$$H_A : \quad \exists \rho^* \in (0,1) \quad \text{s.t.} \quad \begin{cases} y_t \mid x_t \sim F^0_{Y|X=x_t}, & t = 1, 2, \cdots, \tau^* \\ y_t \mid x_t \sim F^1_{Y|X=x_t}, & t = \tau^* + 1, \cdots, n \end{cases}$$

where $\tau^* = \lceil n\rho^* \rceil$. Here $\lceil a \rceil$ denotes the least integer great than or equal to $a$.

- (Localization) When rejecting $H_0$, obtain an estimator $\hat{\tau}$ of $\tau^*$.

**Task II: Analyzing changes in conditional expectations.** We start by considering a simpler task. Assume that $\mathcal{Y} \subset \mathbb{R}$ (i.e., response $y$ is a scalar), $(\mathcal{X}, d)$ is a Euclidean space, and the focus is only on potential changes of the conditional *mean* of $y_t$ given $x_t$. For this simpler task, suppose $f_0, f_1 : \mathcal{X} \to \mathbb{R}$ are two unknown functions with $f_0 \neq f_1$, $\epsilon_t$'s are independent random variables with $\mathbb{E}(\epsilon_t) = 0$ and $\text{Var}(\epsilon_t) < \infty$, and the $\epsilon_t$'s are independent from $x_t$'s. Let $F^0_\epsilon, F^1_\epsilon$ be two probability distributions (not necessarily equal). We are concerned with the following two questions:

- (Detection) Testing the null hypothesis

$$H_0 : \quad y_t = f_0(x_t) + \epsilon_t, \quad \epsilon_t \sim F^0_\epsilon, \quad t = 1, 2, \cdots, n \tag{1}$$

against the alternative

$$H_A : \quad \exists \rho^* \in (0,1) \text{ s.t. } \begin{cases} y_t = f_0(x_t) + \epsilon_t, & \epsilon_t \sim F^0_\epsilon, \quad t = 1, 2, \cdots, \tau^* \\ y_t = f_1(x_t) + \epsilon_t, & \epsilon_t \sim F^1_\epsilon, \quad t = \tau^* + 1, \cdots, n \end{cases} \tag{2}$$

where $\tau^* = \lceil n\rho^* \rceil$.

- (Localization) When rejecting $H_0$, obtain an estimator $\hat{\tau}$ of $\tau^*$.

We will adapt methods developed for Task II to solve Task I. We start with some notations.

**Notations.** Denote $\xrightarrow{d}$ as convergence in distribution. For a sequence of real-valued random variables $X_n$ and a sequence of real numbers $u_n$, denote $X_n = O_{\text{a.s.}}(u_n)$ if $\mathbb{P}(\exists C < \infty, \exists n, \forall m > n, |X_m| \leq Cu_m) = 1$, and $X_n = o_{\text{a.s.}}(u_n)$ if $\mathbb{P}(\forall C > 0, \exists n, \forall m > n, |X_m| \leq Cu_n) = 1$. Denote $\mathbf{0}_p, \mathbf{1}_p \in \mathbb{R}^p$ as the vector of all 0's and 1's, respectively. Denote $I_p \in \mathbb{R}^{p \times p}$ as the identity matrix and $\mathbb{I}$ as the indicator function. Denote $\| \cdot \|_2$ as Euclidean norm.

## 4 Methodology

### 4.1 Solution to Task II

The underlying idea is to transform the CP problem with *paired* data into the more amenable CP problem for *univariate* observations. The best performing transformation, in our view, is reported next. Another possible transformation that has good performance, but without theoretical guarantees, is reported in Appendix A.

The proposed transformation relies on proper estimates of the conditional expectation; specifically, we consider the Nadaraya-Watson (NW) estimator:

$$\hat{f}(\cdot) = \frac{\sum_{t=1}^n k_X \left( h_X^{-1} d(\cdot, x_t) \right) y_t}{\sum_{t=1}^n k_X \left( h_X^{-1} d(\cdot, x_t) \right)}, \tag{3}$$

with $k_X$ a kernel function, and $h_X = h_n$ a sequence satisfying $h_n \to 0$ as $n \to \infty$.

**Introducing the transformed sequence $\tilde{\Delta}_t$.** A natural direction is to find a sequence $\{\hat{\Delta}_t, t = 1, 2, \cdots, n\}$ such that $\hat{\Delta}_t \in [0, \infty)$ represents the degree of difference in conditional expectation for data before $t$ and data after $t$. One such sequence is shown in Figure 1, where we see that without a

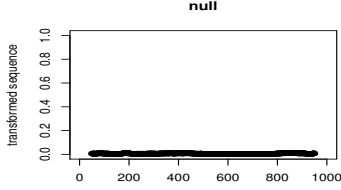
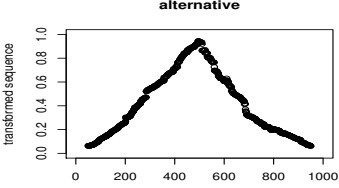

Figure 1: Plot of transformed sequence $\hat{\Delta}_t$ against $t$. In both panels $F_X^t = F_\epsilon^0 = N(0,1)$, $f_0(x) = x$, $n = 1000$. For the right panel $f_1(x) = x^2$, $F_\epsilon^1 = N(0,1)$, $\tau^* = 500$. Y-axis in both panels are set to the same scale for comparison.

change point (left panel), $\hat{\Delta}_t$ is flat across all $t$'s; and with a change point (right panel), $\max_t \hat{\Delta}_t$ will be larger and the maximizer $\arg\max_t \hat{\Delta}_t$ roughly lies around $t = \tau^*$. Thus, inference for the CP problem could be based on the location and magnitude of the maximum among $\{\hat{\Delta}_t\}$.

To define the sequence $\{\hat{\Delta}_t\}$ used for inference (and for generating Figure 1), we introduce a natural sequence, $\{\tilde{\Delta}_t\}$, as follows. At each $t$, consider dividing the data into two parts: before $t$ (i.e., $\{(x_i, y_i), i = 1, 2, \cdots, t\}$) and after $t$ (i.e., $\{(x_i, y_i), i = t+1, t+2, \cdots, n\}$). We fit two functions,

$$\text{the left-side estimate } \hat{f}_-(t, \cdot), \quad \text{using only data } \{(x_i, y_i), i = 1, 2, \cdots, t\}, \quad \text{and}$$
$$\text{the right-side estimate } \hat{f}_+(t, \cdot), \quad \text{using only data } \{(x_i, y_i), i = t+1, t+2, \cdots, n\}. \tag{4}$$

Intuitively, with some mild conditions, under the null, we have for all $t$'s

$$\hat{f}_-(t, \cdot) \approx f_0(\cdot), \quad \hat{f}_+(t, \cdot) \approx f_0(\cdot), \quad \hat{f}_-(t, \cdot) - \hat{f}_+(t, \cdot) \approx 0,$$

while under the alternative, when $t = \tau^*$,

$$\hat{f}_-(t, \cdot) \approx f_0(\cdot), \quad \hat{f}_+(t, \cdot) \approx f_1(\cdot), \quad \hat{f}_-(t, \cdot) - \hat{f}_+(t, \cdot) \approx f_0(\cdot) - f_1(\cdot).$$

However, under the alternative, when $t \neq \tau^*$, both $\hat{f}_+(t, \cdot)$ and $\hat{f}_-(t, \cdot)$ will roughly lie *between* $f_0$ and $f_1$, and we expect the 'difference' between $\hat{f}_-(t, \cdot)$ and $\hat{f}_+(t, \cdot)$ to be smaller than that between $f_0$ and $f_1$. Formally, we define the 'difference' as

$$\mathbb{E}_{X \sim F_X}[\hat{f}_-(t, X) - \hat{f}_+(t, X)]^2, \tag{5}$$

which is different from the usual $l_1$ or $l_2$ distance between functions by emphasizing on their difference at the value of $x$'s with a larger density in $F_X$. In practice we do not know $F_X$, and the expectation in (5) can be approximated by

$$n^{-1}\sum_{i=1}^{n}[\hat{f}_-(t, x_i) - \hat{f}_+(t, x_i)]^2 =: \tilde{\Delta}_t, \tag{6}$$

or more economically, a Monte Carlo estimate using fewer particles ($n' < n$)

$$(n')^{-1}\sum_{t=1}^{n'}[\hat{f}_-(t, x'_t) - \hat{f}_+(t, x'_t)]^2, \tag{7}$$

where $x'_t \overset{\text{iid}}{\sim} \hat{F}_X$ and $\hat{F}_X = (1/n)\sum_{i=1}^{n}\delta_{x_i}$ with $\delta_x$ the delta measure centered at $x$.

**Standardizing $\tilde{\Delta}_t$ into $\hat{\Delta}_t$.** It can be shown that the sequence $\tilde{\Delta}_t$ satisfies the trend shown in Figure 1 (see Theorem C.1 in the Appendix), i.e., when sample size is sufficiently large, $\tilde{\Delta}_t \approx 0$ under the null, while under the alternative, $\tilde{\Delta}_t$ increases with $t$ before change point, while decreases after it. While directly using $\tilde{\Delta}_t$ for solving CP problems might work, a better choice for localization is to consider a properly standardized version (denoted, $\hat{\Delta}_t$) which is comparable across all $t$'s under the null. This requires a more careful analysis of the limiting distribution of $\hat{f}_-, \hat{f}_+$ under the null. Proposition C.2 in the Appendix shows that under the null, with some mild assumptions, for any $t = \lceil n\rho \rceil$ with $\rho \in [\rho_0, \rho_1]$ and any fixed $x$, as $n \to \infty$,

$$\sqrt{n}c_1(h_X)[\hat{f}_-(t, x) - \hat{f}_+(t, x)] \overset{d}{\to} N\left(0, c_2(x)/[\rho(1-\rho)]\right), \tag{8}$$

where $c_1(h_X), c_2(x)$ are constants depending on $h_X$ and $x$ separately. With Definition (6), result (8) immediately suggests setting

$$\hat{\Delta}_t = [t(n-t)/n]\tilde{\Delta}_t, \tag{9}$$

such that $\hat{\Delta}_t$ follows the same limiting distribution (and thus is comparable) for all $t$'s. Notice that the standardized sequence $\{\hat{\Delta}_t, t = 1, 2, \cdots, n\}$ is used in drawing Figure 1. The proposed method is named KCE (Kernel-based change point analysis for Conditional Expectations).

Before summarizing KCE and introducing the detailed algorithm, we would like to discuss the similarity of of the proposed method with the well established CUSUM [35], as the definition (5) and standardization in (9) might look very similar to those in CUSUM. Indeed, one can think of KCE as a generalization of CUSUM where the estimate for mean is replaced by the estimate for conditional expectations; a similar argument holds for the KCD algorithm proposed next to detect changes in conditional distributions.

**Summary of algorithm and complexity.** As mentioned before, after calculating $\hat{\Delta}_t$, the estimator for $\tau^*$ is set to $\hat{\tau} = \arg\max_{n_0 \leq t \leq n_1} \hat{\Delta}_t$. Here $n_0 = \lceil n\rho_0 \rceil$, $n_1 = \lceil n\rho_1 \rceil$, $0 < \rho_0 \leq \rho^* \leq \rho_1 < 1$ are pre-determined in order to prevent selecting too few samples near the ends [8, 9]. We have not discussed yet the detection step. Traditionally, either analytic formulas or re-sampling methods have been used to calculate p-values for CP detection. The (limiting) distribution of $\max \hat{\Delta}_t$ is complicated, and we adopt a resampling-based method to obtain p-values. The complete procedure to solve Task II is summarized in Algorithm 1. Note that we store some pre-computed matrices for speed-up, and the overall time complexity is $O(n^2)$ and space complexity $O(n^2)$.

---

**Algorithm 1** KCE to solve task II (conditional expectation change)

---

    **input:** observations $\{(x_t, y_t)\}_{t=1}^n$, significance level $\alpha$, parameters $n_0, n_1$.
    **output:** estimated change point location $\hat{\tau}$. ($\hat{\tau} = n$ implies no significant change point)
    **pre-compute:**
    1. $K_X = [k_X \left( h_X^{-1} d(x_i, x_j) \right)]_{i,j=1}^n \in \mathbb{R}^{n \times n}$.
    2. $A \in \mathbb{R}^{n \times n}$, where $A_{ij} = \sum_{l=1}^j [K_X]_{il}$.
    3. $B \in \mathbb{R}^{n \times n}$, where $B_{ij} = [K_X]_{ij} y_j$.
    4. $C$, where $[C]_{ij} = \sum_{l=1}^j B_{il}$.
    **for** $t = n_0, n_0 + 1, \cdots, n_1$ **do**
        **for** $i = 1, 2, \cdots, n$ **do**
            calculate $\hat{f}_-(x_i, t) = [C]_{it}/A_{it}$ and $\hat{f}_+(x_i, t) = ([C]_{in} - [C]_{it})/(A_{in} - A_{it})$.
        **end for**
        calculate $\hat{\Delta}_t = [t(n - t)/n] \sum_{i=1}^n [\hat{f}_-(t, x_i) - \hat{f}_+(t, x_i)]^2$.
    **end for**
    **detection:** obtain p-value for $\max_{n_0 \leq t \leq n_1} \hat{\Delta}_t$ using permutations or bootstrap.
    **localization:** if p-value $< \alpha$, estimate $\hat{\tau} = \arg\max_{n_0 \leq t \leq n_1} \hat{\Delta}_t$; else, set $\hat{\tau} = n$.

---

## 4.2 Solution to Task I

So far we have derived the solution for Task II, the conditional expectation change point problem. We show next how to adapt the proposed procedure to solve the more difficult Task I, the conditional distribution change point problem.

**Generalizing $\tilde{\Delta}_t$.** Note that the NW estimator is linear on $y_t$'s, and thus $\tilde{\Delta}_t$ can be written as

$$\sum_{i,j=1}^t w(t,i,j,x_i,x_j)y_iy_j + \sum_{i,j=t+1}^n w(t,i,j,x_i,x_j)y_iy_j - 2\sum_{i=1}^t \sum_{j=t+1}^n w(t,i,j,x_i,x_j)y_iy_j, \quad (10)$$

where denote for simplicity $k_X \left( h_X^{-1} d(x_i, x_j) \right) = k_X(i, j)$, then

$$w(t,i,j,x_i,x_j) = \begin{cases} \frac{1}{n}\sum_{l=1}^n \frac{k_X(i,l)}{\sum_{r=1}^t k_X(i,r)} \frac{k_X(j,l)}{\sum_{r=1}^t k_X(j,r)}, & \text{if } i,j \leq t, \\ \frac{1}{n}\sum_{l=1}^n \frac{k_X(i,l)}{\sum_{r=t+1}^n k_X(i,r)} \frac{k_X(j,l)}{\sum_{r=t+1}^n k_X(j,r)}, & \text{if } i,j > t, \\ \frac{1}{n}\sum_{l=1}^n \frac{k_X(i,l)}{\sum_{r=1}^t k_X(i,r)} \frac{k_X(j,l)}{\sum_{r=t+1}^n k_X(j,r)}, & \text{if } i \leq t, j > t. \end{cases} \quad (11)$$

We replace the inner product terms $y_i y_j$ in Equation (10) by $k_Y(y_i, y_j)$ (a standard kernel technique), yielding

$$\tilde{\Delta}_t = \sum_{i,j=1}^{t} w(t,i,j,x_i,x_j) k_Y(y_i,y_j) + \sum_{i,j=t+1}^{n} w(t,i,j,x_i,x_j) k_Y(y_i,y_j)$$
$$- 2 \sum_{i=1}^{t} \sum_{j=t+1}^{n} w(t,i,j,x_i,x_j) k_Y(y_i,y_j). \tag{12}$$

Some notes on (12): if $x_t \equiv c$ for all $t$'s (which can be viewed as the canonical *unpaired* setting, observing only $\{y_t\}_{t=1}^{n}$), $\tilde{\Delta}_t$ equals the maximum mean discrepancy [16] for testing equality between the distribution of $\{y_i, i \leq t\}$ and that of $\{y_i, i > t\}$. In the *paired* setting, the weights (11) depend also on the value of $x_t$'s: the more similar $x_i, x_j$ are, the larger the weight on $k_Y(y_i, y_j)$ becomes.

**Standardizing** $\tilde{\Delta}_t$**.** Similar to Task II, one can show under some mild assumptions that $\tilde{\Delta}_t$ defined in (12) has the same trend as in Figure 1 (see Theorem 5.1 in Section 5). Standardizing $\tilde{\Delta}_t$, however, is much less obvious. We defer the rigorous derivation to Theorem 5.1 in the following section. Here we give some high-level intuition: standardization essentially relies on variance of $\tilde{\Delta}_t$; when deriving it, because of the randomness of $x_i$'s, a uniform bound that does not depend on $x_i$'s is needed. To establish uniform bounds, we need to put some assumptions on the space of $x$'s. In order to regulate the complexity of the space $\mathcal{X}$, we introduce Kolmogorov's entropy, defined as:

$$\psi_{\mathcal{X}}(\epsilon) = \log(N_\epsilon(\mathcal{X})),$$

with $N_\epsilon(\mathcal{X})$ the minimal number of open balls of radius $\epsilon$ in order to cover $\mathcal{X}$. Now we are ready to present the standardization for $\tilde{\Delta}_t$. Recall the re-scaling factor in (9) is $1/t + 1/(n-t)$, and we expect the re-scaling factor here to be inflated by the Kolmogorov entropy of space $\mathcal{X}$ in some way. The simplest case is when $\mathcal{X}$ is compact and $X$ has a density with respect to the Lebesgue measure: we can still set $\hat{\Delta}_t$ using definition (9); and in the general case we suggest setting

$$\hat{\Delta}_t = [(1/t)\psi_{\mathcal{X}}((\log t)/t) + (1/(n-t))\psi_{\mathcal{X}}((\log(n-t))/(n-t))]^{-1} \tilde{\Delta}_t.$$

Intuitively, for any fixed $t = \lceil n\rho \rceil$, the larger the sample size, the smaller the variance of $\tilde{\Delta}_t$, and thus the smaller the rescaling factor. For any fixed $n$ and $t$, the more complex the space $\mathcal{X}$, the larger $\psi_{\mathcal{X}}$, and the larger the rescaling factor. A more detailed derivation on this standardization is included in the next section.

**Summary of algorithm and complexity.** The complete procedure for solving task I is summarized in Algorithm 2. Again we store some pre-computed matrices to save time. The time complexity of Algorithm 2 is $O(n^3)$ and space complexity $O(n^2)$. The proposed method is named KCD (Kernel-based change point analysis for Conditional Distributions).

# 5   Theory

We present only some relevant theory for Task I, and defer the rest to Section C and all proofs to Section D in the Appendix. From the Moore-Aronszajn theorem, the function $k_Y(y, \cdot)$ satisfies $k_Y(y, y') = \langle k_Y(y, \cdot), k_Y(y', \cdot) \rangle_{\mathcal{H}}$ where $\mathcal{H}$ is a reproducing kernel Hilbert space, as long as kernel $k_Y(\cdot, \cdot)$ is positive-definite. Denote the conditional mean maps before and after change point as

$$f_0(x) = \mathbb{E}_{Y \sim F_{Y|X=x}^0}[k_Y(Y, \cdot)], \quad f_1(x) = \mathbb{E}_{Y \sim F_{Y|X=x}^1}[k_Y(Y, \cdot)],$$

in the sense that $\langle f_i(x), k_Y(y, \cdot) \rangle_{\mathcal{H}} = \mathbb{E}_{Y \sim F_{Y|X=x}^i}[k_Y(Y, y)]$ for $i = 0, 1$. Define the norm $\| \cdot \|_{\mathcal{H}} = \sqrt{\langle \cdot, \cdot \rangle_{\mathcal{H}}}$, and let

$$\Delta = \mathbb{E}_{X \sim F_X} \| f_0(X) - f_1(X) \|_{\mathcal{H}}^2. \tag{13}$$

Let $B(x, h) = \{x' \in \mathcal{X}, d(x', x) \leq h\}$ be a closed ball in $\mathcal{X}$. We need the following assumptions to establish the asymptotic behavior of $\tilde{\Delta}_t$.

**Assumption 1** (small ball probability of $F_X$)**.** *There exists a non-decreasing function $m(\cdot)$ such that*

$$\exists (C_1, C_2), \forall x \in \mathcal{X}, \forall \epsilon > 0, 0 < C_1 m(\epsilon) \leq \mathbb{P}(X \in B(x, \epsilon)) \leq C_2 m(\epsilon) < \infty.$$

**Algorithm 2** KCD to solve Task I (conditional distribution change)

---

**input:** observations $\{(x_t, y_t)\}_{t=1}^n$, significance level $\alpha$, parameters $n_0, n_1$.
**output:** estimated change point location $\hat{\tau}$. ($\hat{\tau} = n$ implies no significant change point)
**pre-compute:**
1. $K_X = [k_X(h_X^{-1}d(x_i, x_j))]_{i,j=1}^n \in \mathbb{R}^{n \times n}$, $K_Y = [k_Y(y_i, y_j)]_{i,j=1}^n \in \mathbb{R}^{n \times n}$.
2. $A \in \mathbb{R}^{n \times n}$, where $A_{ij} = \sum_{l=1}^j [K_X]_{il}$.
3. $B = (1/n)K_X K_X^\top \in \mathbb{R}^{n \times n}$.
4. $C \in \mathbb{R}^{n \times n}$, where $C_{ij} = B_{ij}[K_Y]_{ij}$.
**for** $t = n_0, n_0 + 1, \cdots, n_1$ **do**
    calculate $Q \in \mathbb{R}^{n \times n}$ with

$$Q_{ij} = \begin{cases} \frac{C_{ij}}{A_{it}A_{jt}}, & \text{if } i, j \leq t, \\ \frac{C_{ij}}{A_{it}(A_{jn} - A_{jt})}, & \text{if } i \leq t, j > t, \\ \frac{C_{ij}}{(A_{in} - A_{it})(A_{jn} - A_{jt})}, & \text{if } i, j > t. \end{cases}$$

    calculate $\tilde{\Delta}_t = \sum_{i,j=1}^t Q_{ij} + \sum_{i,j=t+1}^n Q_{ij} - 2\sum_{i=1}^t \sum_{j=t+1}^n Q_{ij}$.
**end for**
**detection:** obtain $p$-value for $\max_{n_0 \leq t \leq n_1} \hat{\Delta}_t$ using permutations or bootstrap.
**localization:** if $p$-value $< \alpha$, estimate $\hat{\tau} = \arg\max_{n_0 \leq t \leq n_1} \hat{\Delta}_t$; else, estimate $\hat{\tau} = n$.

---

**Assumption 2** (Lipschitz continuity of $f_0, f_1$). *Functions $f_0, f_1$ satisfy*

$$\exists C_3 < \infty, \exists b > 0, \forall x, x' \in \mathcal{X}, \|f_0(x) - f_0(x')\|_{\mathcal{H}} + \|f_1(x) - f_1(x')\|_{\mathcal{H}} \leq C_3 d^b(x, x').$$

**Assumption 3** (regularization on kernel $k_X$). *$k_X(\cdot)$ is a nonnegative, bounded and Lipschitz continuous function with support $[0, 1)$, and if $k_X(1) = 0$ the following conditions have to be satisfied*

$$\exists (C_4, C_5), \ -\infty < C_4 < k_X'(t) < C_5 < 0, \quad \forall t \in [0, 1),$$
$$\exists C_6 > 0, \exists \eta_0 > 0, \forall \eta < \eta_0, \int_0^\eta m(u)du > C_6 \eta m(\eta).$$

**Assumption 4** (regularization on kernel $k_Y$). *$k_Y(\cdot, \cdot)$ is a continuous, symmetric, positive definite, and uniformly bounded kernel.*

**Assumption 5** (topological complexity of $\mathcal{X}$). *The topological complexity $\psi_{\mathcal{X}}(\cdot)$, together with the small ball probability function $m(\cdot)$, satisfy*

$$\exists C_7 > 0, \exists \eta_0 > 0, \forall \eta < \eta_0, 0 \leq m'(\eta) < C_7,$$
$$\exists n_0, \forall n > n_0, (\log n)^2/[nm(h)] < \psi_{\mathcal{X}}(\log n/[nm(h)]) < nm(h)/[\log n],$$
$$\exists \beta > 1, \sum_{n=1}^\infty \exp\{(1-\beta)\psi_{\mathcal{X}}((\log n)/n)\} < \infty.$$

**Remark 5.1.** *Assumptions 1-5 are technical, but, in general, they are mild and not that restrictive. For example, when the space $\mathcal{X}$ is compact, and $k_X, k_Y$ are Gaussian kernels, Assumption 1, 3, 4 are all satisfied. If we further assume $\mathcal{X}$ is Euclidean and the random variable $x$ has a density function, Assumption 5 is satisfied. Assumption 2 is probably the most abstract, but it essentially states that the conditional distribution $p(Y \mid X = x)$ changes smoothly in the sense that $p(Y \mid X = x_0)$ is close to $p(Y \mid X = x_1)$ when $x_0$ is close to $x_1$.*

**Theorem 5.1.** *Suppose Assumptions 1, 2, 3, 4, 5 hold.*

*(1) Under the null, for any $t = \lceil n\rho \rceil$ with $\rho \in [\rho_0, \rho_1]$,*

$$\tilde{\Delta}_t = O_{a.s.}\left([tm(h)]^{-1}\psi_{\mathcal{X}}(t^{-1}\log t) + [(n-t)m(h)]^{-1}\psi_{\mathcal{X}}((n-t)^{-1}\log(n-t))\right).$$

*(2) Under the alternative, for any $t = \lceil n\rho \rceil$ with $\rho \in [\rho_0, \rho^*) \cap (\rho^*, \rho_1]$,*

$$\tilde{\Delta}_t - \delta(\rho)\Delta = o_{a.s.}(1), \quad \text{where} \quad \delta(\rho) = \begin{cases} (1-\rho^*)^2/(1-\rho)^2, & \text{if } \rho \leq \rho^* \\ (\rho^*)^2/\rho^2, & \text{if } \rho > \rho^* \end{cases}. \tag{14}$$

**Remark 5.2.** *Assumptions 1, 2, 3, 4 are standard conditions for obtaining pointwise convergence of NW estimators. Assumption 5 comes from [13] and regulates the topological structure of the infinite dimensional space $\mathcal{X}$. An example of $m, \psi_{\mathcal{X}}$: if $\mathcal{X}$ is a compact subset of $\mathbb{R}^p$, $\psi_{\mathcal{X}}(\epsilon) = O(\log(1/\epsilon))$, and, if $X$ has a density with respect to the Lebesgue measure, $m(h) = O(h^p)$.*

Table 1: Mean $\pm$ standard error of the $l_1$ localization error $|\hat{\tau} - \tau^*|$ for each method, summarized over 20 simulations. The best performing method is marked in bold font.

(a) Experiment A.

| $f_1(x)$ | [31] | $D_Y$ | $D_{XY}$ | KCE |
|---|---|---|---|---|
| $5x$ | 98.4±10.6 | 322.5±55.6 | **1.7±0.3** | 2.1±0.3 |
| $\cos(x)$ | 239.1±41.1 | 320.6±48.4 | 6.4±1.6 | **2.7±0.5** |
| $x^2$ | 70.2±28.8 | 285.2±44.2 | 9.4±1.9 | **4.9±1.3** |
| $|x|$ | 240.1±47.3 | 297.7±46.1 | 11.8±6.4 | **2.9±0.5** |
| $0.1\max(0, 1-x)$ | 311.7±44.3 | 248.8±53.1 | 10.2±4.3 | **6.4±1.6** |
| $e^x$ | 56.6±22.9 | 257.4±41.2 | 3.6±1.2 | **1.9±0.4** |
| $\frac{1}{2(x+3)}$ | 307.8±40.5 | 369.6±53.1 | 10.9±2.9 | **3.1±0.6** |

(b) Experiment B.

| $v_1(x)$ | [31] | $D_Y$ | $D_{XY}$ | KCD |
|---|---|---|---|---|
| $10x$ | 89.0±12.6 | 2.7±0.5 | 3.6±0.7 | **1.3±0.2** |
| $\cos(x)$ | 284.7±44.0 | 146.8±43.7 | 4.8±1.5 | **3.0±0.9** |
| $x^2$ | 107.9±25.2 | 46.1±13.7 | 38.4±11.7 | **12.6±3.0** |
| $\max(0, 1-x)$ | 107.6±11.7 | 1.9±0.3 | 2.6±0.6 | **1.8±0.3** |
| $e^x$ | 97.5±12.0 | 63.0±14.0 | 6.6±1.7 | **3.1±0.8** |
| $\frac{1}{x+3}$ | 249.2±34.7 | 10.4±3.9 | 13.9±3.8 | **5.0±1.3** |

**Remark 5.3.** *Theorem 5.1 shows that as long as $\Delta \neq 0$, $\hat{\Delta}_t$ satisfies the trend in Figure 1. Thus, performance of KCD is crucially affected by $\Delta$, which is characterized in the next proposition.*

**Proposition 1.** *Under Assumption 2, if $k_Y$ is a characteristic kernel, and $F^0_{Y|X=x} \neq F^1_{Y|X=x}$ for at least one $x \in supp(F_X)$, i.e., the support of $F_X$, we have $\Delta \neq 0$.*

**Remark 5.4.** *Proposition 1 shows that $\Delta$ acts as a discrepancy measure between two sets of conditional distributions, $\{F^0_{Y|X=x}, x \in supp(F_X)\}$ and $\{F^1_{Y|X=x}, x \in supp(F_X)\}$. It is reminiscent of the maximal conditional mean discrepancy (MCMD) between two conditional distributions in [36]. Actually, $\Delta$ equals the expected value of squared MCMD between $F^0_{Y|X}, F^1_{Y|X}$ with $X \sim F_X$. A detailed discussion on the link with [36] is included in the Appendix (Section B).*

## 6 Experimental Results

This section investigates performance in synthetic data. We report representative results on different forms of $\mathcal{X}, \mathcal{Y}$ and types of changes, with additional results included in the Appendix.

**Baselines.** We consider three baselines: one existing (the fixed design CP method [31]), and two adapted from existing abrupt CP methods for unpaired data (denoted by $D_{XY}$ and $D_Y$). $D_{XY}$ applies an existing CP method by treating $(x_t, y_t)$ as the observation, while $D_Y$ does so by discarding $x_t$. There are numerous CP methods to choose from, and we use $S_1$ introduced in [33], which unifies many existing nonparametric CP methods and can be viewed exactly as Algorithm 2 by setting $x_t \equiv c$. For KCD (or KCE), $D_{XY}, D_Y$, we set $k_X(u) = \exp\{-u^2\}$ and the choice of $k_Y$ depends on $\mathcal{Y}$. For the method in [31], there are restrictions on the kernel function that are violated by the RBF kernel. So, following [31], we set $k(u) = 1.5(1 - u^2)\mathbb{I}(0 \leq u \leq 1)$. All bandwidths used for all methods are tuned among $S_h = \{0.001, 0.01, 0.1, 1, 10\}$ on 10 independently generated data sets.

**Evaluation metrics.** We report the $l_1$ localization error $|\hat{\tau} - \tau^*|$ and the empirical power, both averaged over 20 independent simulations. Power is calculated under significance level $\alpha = 0.05$ with p-value determined by 500 bootstraps.

**Experiment A:** $y \in \mathbb{R}$, $\mathbb{E}[y \mid x]$ **changes.** We first consider task II, and we set $F_X = N(0, 1)$, $F^0_\epsilon = F^1_\epsilon = N(0, 1)$, $n = 1000$, $\rho^* = 0.7$, $\tau^* = 700$, $\rho_0 = 0.05$, $\rho_1 = 0.95$. Before the change point, we set $f_0(x) = x$, while after change point, we investigate different options for $f_1$. For $D_Y$, set $k_Y(y, y') = \exp\{-\|y - y'\|_2^2/h_Y^2\}$; for $D_{XY}$, set $k_Z(z, z') = \exp\{-\|z - z'\|_2^2/h_Z^2\}$ with $z_t = (x_t, y_t)'$. KCE uses Algorithm 1 with $d(\cdot, \cdot)$ set to Euclidean distance.

Localization comparisons are reported in Table 1a. Note that the method in [31] and $D_Y$ perform not so well, which is expected as the former is designed for equi-spaced covariates, and the latter only looks at the marginal distribution of $y_t$; $D_{XY}$ performs better, but is still slightly worse than KCE. KCE performs quite well, obtaining the most accurate estimator in most settings. Power comparisons are similar and are included in Table 3a in the Appendix.

**Experiment B:** $y \in \mathbb{R}$, $p(y \mid x)$ **changes.** Here we investigate task I. We start with $y \in \mathbb{R}$, $\mathbb{E}[y \mid x]$ does not change, yet $p(y \mid x)$ changes. Specifically, we consider the model

$$y_t = \begin{cases} v_0(x_t)\epsilon_t, & t \leq \tau^*, \\ v_1(x_t)\epsilon_t, & t > \tau^*, \end{cases}$$

where $\epsilon_t \overset{iid}{\sim} N(0, 1)$, $v_0(x) = x$ and we consider different forms of $v_1$'s. We set $F_X = N(0, 1)$, $n = 1000$, $\rho^* = 0.7$, $\tau^* = 700$, $\rho_0 = 0.05$, $\rho_1 = 0.95$. KCD uses Algorithm 2 with $k_Y(y, y') =$

$\exp\{-\|y - y'\|_2^2/h_Y^2\}$ and $d(\cdot, \cdot)$ set to Euclidean distance. $D_Y$ uses $k_Y(y, y') = \exp\{-\|y - y'\|_2^2/h_Y^2\}$ and $D_{XY}$ uses $k_Z(z, z') = \exp\{-\|z - z'\|_2^2/h_Z^2\}$ with $z_t = (x_t, y_t)'$.

Localization results are summarized in Table 1b. We observe that KCD performs the best among all alternatives, followed by $D_{XY}$, $D_Y$, and lastly [31].

**Experiment C: general** $y \in \mathcal{Y}$, $p(y \mid x)$ **changes.** In the most general setting, let us consider task I with $x \in \mathbb{R}^p$ with $p = 5$, $F_X = N(\mathbf{0}_p, I_p)$ and different forms of $\mathcal{Y}$'s. When designing experiments, we deliberately let the conditional mean of $y$ given $x$ to be invariant. We set $k_Z(z, z') = k_X(h_X^{-1}d(x, x'))k_Y(y, y')$ for $D_{XY}$, and $k_Y$ used in all methods is tailored to each $\mathcal{Y}$. [31] is not applicable to non-scalar $y_t$'s and is thus omitted. The other settings are identical to Experiment B. The forms of $\mathcal{Y}$'s that are considered include:

(1) $\mathcal{Y} = \mathbb{R}^p$, and we set $y_t \sim N(\mathbf{0}_p, I_p)$ before change point, and $y_t \sim N(B'x_t, I_p)$ with $B = [\beta_1, \beta_2, \cdots, \beta_5]$ and $\beta_i = 0.1 \times i \times \lambda \times \mathbf{1}_p$ with different magnitude of $\lambda$ after the change point. We set $k_Y(y, y') = \exp\{-\|y - y'\|_2^2/h_Y^2\}$.

(2) $\mathcal{Y} = \mathbb{R}^{p \times p}$. Denote $Y_{ij}$ as the element in the $i$-th row and $j$-th column of $Y$. Before change point, we set $Y_{ij} = X_i X_j$ where $X_i, X_j$ denote the element in the $i$th and $j$th coordinate of $X$, while after change point, we set it to a different quantity. We set $k_Y(Y, Y') = \exp\{-\|Y - Y'\|_F^2/h_Y^2\}$ with Frobenius norm $\|\cdot\|_F$.

(3) $\mathcal{Y} = \mathcal{P}$, the set of all univariate normal distributions with variance 1. Each datum $y_t$ is $m = 10$ observations independently drawn from $N(\mu(x_t), 1)$. We consider $\mu(x_t) = (\lambda/5)x_t'\mathbf{1}_5$ with different magnitude $\lambda$'s. We set $k_Y(Y, Y') = \exp\{-\|Y - Y'\|_W^2/h_Y^2\}$ with 2-Wasserstein metric $\|\cdot\|_W$.

Table 2: Mean $\pm$ standard error of the $l_1$ localization error $|\hat{\tau} - \tau^*|$ for each method, summarized over 20 simulations in experiment C. The best performing method is marked in bold font.

| $\mathcal{Y}$ | $H_A$ | $D_Y$ | $D_{XY}$ | KCD |
|---|---|---|---|---|
| $\mathbb{R}^p$ | $\lambda = 1.0$ | 7.1±2.0 | 15.1±4.4 | **6.7±2.1** |
| $\mathbb{R}^p$ | $\lambda = 0.8$ | 11.7±3.9 | **9.4±3.2** | 11.6±3.3 |
| $\mathbb{R}^p$ | $\lambda = 0.6$ | 65.1±22.6 | 69.8±28.7 | **38.2±12.9** |
| $\mathbb{R}^{p \times p}$ | $Y_{ij} = X_i(X_j)^3$ | 10.4±2.8 | 10.2±2.8 | **5.7±1.4** |
| $\mathbb{R}^{p \times p}$ | $Y_{ij} = (X_i)^3(X_j)^3$ | **10.2±3.3** | 15.5±4.1 | **10.2±2.9** |
| $\mathbb{R}^{p \times p}$ | $Y_{ij} = \sin(X_i)\sin(X_j)$ | 1.9±0.7 | 5.9±1.4 | **1.4±0.6** |
| $\mathcal{P}$ | $\lambda = 1.0$ | 51.6±17.6 | 5.1±1.7 | **4.0±0.8** |
| $\mathcal{P}$ | $\lambda = 0.8$ | 25.2±9.5 | **7.0±1.7** | 9.2±2.3 |
| $\mathcal{P}$ | $\lambda = 0.5$ | 113.7±35.7 | 104.7±31.7 | **49.1±12.3** |

Results for experiment C are summarized in Table 2. In general, KCD (or KCE) outperform $D_{XY}$ and $D_Y$, often with significant improvements. This observation is consistent across all experiments. The trade-off is that KCD takes $O(n^3)$ time complexity, while KCE, $D_{XY}, D_Y$ take only $O(n^2)$.

# 7 Real Data Applications

**UK stock index.** The Financial Times Stock Exchange 100 Index (FTSE 100) is a share index of the 100 largest companies listed on the London Stock Exchange. We investigate potential changes in UK economy from January to August 2016, a period containing the Brexit vote (06/23/2016). Following [20, 21], we consider changes in the daily value of FTSE 100 stock index, compared to two "reference" time series: the NYSE Composite Index and the NIKKEI 225 stock index. The NYSE Composite is a stock market index of all common stocks listed on the New York Stock Exchange, and the NIKKEI 225 is a stock market index for the Tokyo Stock Exchange operating in the Japanese Yen. They are used here as indicators for the US and Japan economies. Following [20], we divide the closing value of each index by its corresponding daily Euro exchange rates. All data are downloaded from https://www.marketwatch.com/. Data from the three stock indices are shown in Figure 2a.

The processed dataset contains $n = 157$ sample points, with $y \in \mathbb{R}$ the FTSE 100 divided by Euro exchange rate to British pound, and $x \in \mathbb{R}^2$ the vector consisting of NYSE and NIKKEI 225 divided by Euro exchange rate to US dollars and Japanese Yen, respectively. Using KCD with $k_X(u) = \exp\{-u^2\}$, $d(x, x') = \|x - x'\|_2$, $h_X^2 = 0.1$, $k_Y(y, y') = \exp\{-\|y - y'\|_2^2/h_Y^2\}$, $h_Y^2 = 0.1$ and $n_0 = 5, n_1 = n - 5$, we identify 06/23/2016 as the change point, with a p-value $< 0.01$. At the significance level 0.05, we conclude that there is a change point, which is consistent with [20, 21], and the estimated change point is also the date of the Brexit vote. https://www.overleaf.com/project/625c4c34c121ad5a0311e7cc

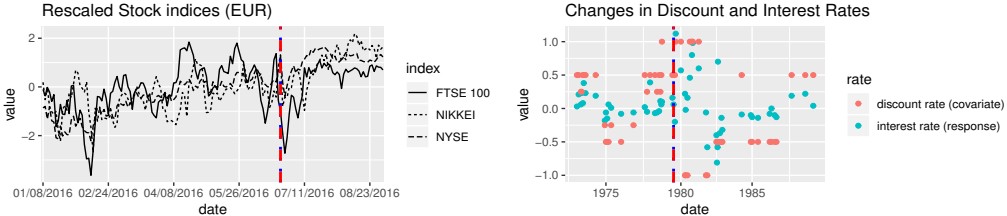

(a) UK stock index example. The red dashed vertical line represents the date of the Brexit vote.

(b) Market interest rates example. The red dashed vertical line represents the CP estimated by [4].

Figure 2: Real data examples. In both panels, the blue dotted line represents $\hat{\tau}$ estimated using KCD.

**Market interest rates.** We consider changes in the relationship between market interest and discount rates over 1973-1989. The discount rate is the rate "at which the Federal Reserve System (Fed) lends and is set by the Fed" [4]. Market interest rates usually respond to changes in discount rates, but the differentials in response might be different, reflecting different conditions of the economy. We examine the response of market interest rates to discount rates over time. The data we use are collected by [10], where the yields of three-month T-bills are treated as market interest rates.

The data set contains $n = 56$ sample points, with covariate $x \in \mathbb{R}$ the discount rate, response $y \in \mathbb{R}$ the market interest rate. We note that discount rates are changed irregularly, thus the sequence is not equi-spaced in terms of time. The sequence $\{(x_i, y_i)\}_{i=1}^n$ is visualized in Figure 2b.

Using KCD with $k_X(u) = \exp\{-u^2\}$, $d(x, x') = \|x - x'\|_2$, $h_X^2 = 0.1$, $k_Y(y, y') = \exp\{-\|y - y'\|_2^2/h_Y^2\}$, $h_Y^2 = 0.1$ and $n_0 = 5, n_1 = n - 5$, we estimate a change point 08/17/1979. This is identical to the estimation in [4], and is close to the October 1979 change in the Fed's operating procedures ([4]). However, the p-value is 0.25, indicating that the change is not significant. This is similar to the result obtained using the vanilla symmetric response model in [4], but different from that using adjusted or asymmetric response model [4].

## 8 Conclusion and Discussion

To identify changes in the conditional distribution of paired sequences, we investigate several approaches, some based on existing algorithms and one novel. Our nonparametric method applies to general covariates and responses, and outperforms existing methods. Future directions of research include relaxing distributional assumptions, and improving the $O(n^3)$ time complexity of Algorithm 2. As suggested by one reviewer, the question of detecting changes in conditional distributions with multiple or gradual change points is also important but unresolved. A rigorous treatment of this generalized setup requires nontrivial modifications of the method and a re-formulation of the mathematical set-up, but we provide a simple windowing approach as a possible solution for multiple change points in the Appendix. There are no foreseeable negative societal impacts of this work.

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
