# OpenReview forum: "Detection and Localization of Changes in Conditional Distributions"
_NeurIPS.cc/2022/Conference — NeurIPS 2022 Accept_

### Official Review · Reviewer_CXB3 · 2022-07-05

**Rating:** 7
**Confidence:** 3
**Soundness:** 4 excellent
**Presentation:** 3 good
**Contribution:** 3 good

**Summary:**

The paper presents a novel approach to detecting change points in conditional distributions, in particular for a setting relevant in machine learning where covariates are drawn iid from a fixed distribution, but the relation to the label changes. The idea is to derive a sequence of differences in conditional distribution, standardize this sequence, and perform a hypothesis test whether a change point exists for the maximum difference in an interval and report his point. For detecting a change in the conditional expectation, the difference of conditional expectation before and after this point is estimated using the kernel-based Nadaraya-Watson estimator. In order to estimate differences in conditional distributions, the paper proposes to generalize the estimator to take into account all pairwise similarities of labels (measured by a suitable kernel). The paper then proves correctness for their approach. The method is empirically evaluated on synthetic data and two real-world use cases.

The paper is sound, the presentation is clear, the method is novel and interesting, and well placed within the literature. I vote for acceptance.

**Questions:**

- regarding the UK Stock experiment: there seems to be a second changepoint coinciding with the Brexit vote announcement in February, 2016. Does your method detect this as well?

**Limitations:**

The method is currently used to detect a single change point in a time series. It would be great to empirically evaluate whether it can also be used to detect multiple change points (e.g., by a windowing approach), in particular on real-world use cases, e.g., detecting stock market bubbles in stock prices. It would also be great to see whether the method is able to detect less abrubt changes in practice (e.g., by shifting the mean gradually in a short time interval from the initial distribution to the final one).

**Strengths And Weaknesses:**

Strengths:
- tackles a problem relevant to the NeurIPS community
- novel and sound method
- solid theoretical analysis

Weakness:
- the real-world use-cases are a bit limited.

---

> ### Author Response · Authors · 2022-08-02
> **Response to Reviewer CXB3**
>
> We thank the reviewer for the encouraging feedback and insightful comments!
>
> **Whether our method detects a second change point around February 2016**
> * No, our method is designed for a single change point setup, so once we find the first change point, we stop. But we agree that generalizing to multiple change points setting is a realistic and interesting direction for future work. As the reviewer mentioned, a sliding window approach is a possibility; and simple binary segmentation is also worth investigating. The theoretical analysis that is needed to guarantee that the proposed method can still work when multiple change points exist is less straightforward and we leave it for future work. We will add some discussion to this generalization in our final version. Thank you for raising this important point!
>
> **Whether the method is able to detect less abrupt changes in practice**
> * This is an interesting question for which we currently do not have an answer.  Note that for gradual changes, defining a proper type of alternative is more difficult. In Task II (conditional mean shift), we agree that we can define the alternative as a smooth function added to the original mean function (which is also the setup considered in some previous literature in this strand, for example, [1]). But for Task I, this is less obvious. We will add some discussion to this question in the final version and leave it as an interesting direction for future work.
>
> **References**
>
> [1] Michael Vogt. Testing for structural change in time-varying nonparametric regression models. 398 Econometric Theory, 31(4):811–859, 2015.

---

> > ### Comment · Reviewer_CXB3 · 2022-08-05
> > **Response to the authors**
> >
> > Dear authors,
> >
> > Thank you for your response and for adding those discussions to the manuscript. I know it is more of a gimmick, but for the real-world experiments you have chosen, multiple change point detection would be a great extension. So please consider adding a simple experiment to the appendix with either windowing or binary segmentation, even though there is no theoretical guarantee for this. To me, it would nicely round up the story for the experiments. However, this is not critical and does not change my evaluation of the paper.
> >
> > Best regards

---

> > > ### Author Response · Authors · 2022-08-05
> > > **Multiple change points**
> > >
> > > Thank you for the suggestion. We will look carefully at multiple points in the real data for the final paper (it is easy to implement a binary segmentation algorithm - one for which we have no theoretical guarantees).

---

### Official Review · Reviewer_cmHY · 2022-07-10

**Rating:** 7
**Confidence:** 5
**Soundness:** 3 good
**Presentation:** 3 good
**Contribution:** 3 good

**Summary:**

The paper discusses a nonparametric change-point estimation methodology for change-point detection in paired sequence of observations. The author(s) propose a methodology based on kernel smoothing estimator to detect change-point in conditional expectation. Further, they generalized the methodology to change-point detection in conditional distribution via a novel kernel estimator. The asymptotic properties of the estimator has been derived. The synthetic results and real data analysis validate the performance of the proposed estimator.

**Questions:**

Following are my queries for the author(s)
(A) In proposition 1 we require $\Delta\neq 0$, is it possible to have an asymptotic requirement i.e. as sample size n goes to infinity $\Delta$ remain bounded above zero
(B) Are we allowed to have discrete covariates x's in the settings author(s) proposed?
(C) How is the empirical power calculated in the experiments in section 6?

**Limitations:**

Yes they adequately addressed the limitations and potential negative societal impact of their work

**Strengths And Weaknesses:**

strengths

(A) A novel change-point detection technique for paired sequence of observations
(B) Smart use of NW eatimator for change in mean detection
(C) novel theoretical results discussing the asymptotic properties of the proposed kernel change-point estimator for change detection in conditional distribution

weaknesses
(A) It wasn't clear in both algorithm 1 and 2 how one should make the choice of h_X or h_Y?
(B) permutation/bootstrap procedure for p-value computation could be computationally quite expensive
(C) standardized version of \hat{Delta}_t is quite close to the CUSUM type statistic applied in change point detection. Some discussion would have been useful there.

---

> ### Author Response · Authors · 2022-08-02
> **Response to Reviewer cmHY**
>
> We thank the reviewer for the positive feedback and constructive comments!
>
> **On the choice of $h_X$ or $h_Y$**
> * As it is common for all kernel methods, the bandwidth is either set a priori or it requires tuning. In our simulations we tune it; in our real data analysis, we subjectively set a value $h_X^2=h_Y^2=0.1$ without tuning and the proposed method still works well.
>
> **On the permutation/bootstrap procedure**
> * The reviewer is correct - permutation/bootstrap procedures can be expensive. In the nonparametric setup of our paper, deriving an accurate analytic formula for p-values can be very difficult, leading to our adoption of this option. Please note that this has been used for some previous change point methods, for example, [1].
>
> **Connection to standardization of $\hat\Delta_t$ and CUSUM**
> * Thank you for the insightful comment! Yes, they indeed share many similarities both from their form and the mathematics hidden behind. We will add more discussion to this point in the final version.
>
> **Is it possible to have an asymptotic requirement i.e. as sample size $n$ goes to infinity $\Delta$ remain bounded above zero?**
> * If we understand correctly, the reviewer is saying that from Theorem 5.1, the proposed method succeeds only when $\Delta\ne 0$; and the reviewer is asking whether it can also work when $\Delta$ going to zero but at a certain rate. We note that the final statistic we use is $\hat\Delta_t=[t(n-t)/n]\tilde\Delta_t$. Our hypothesis is that the proposed method can still work as long as $n\Delta$ goes to infinity, although a strict conclusion will definitely involve more careful analysis.
>
> **Are we allowed to have discrete covariates $x$’s in the settings authors proposed?**
> * Yes, the methodology works for discrete covariates as long as the semi-metric space they lie in satisfies the technical assumptions in Section 5.
>
> **How is the empirical power calculated in the experiments in Section 6?**
> * It is calculated using bootstrap distributions. In more detail: we first calculate $\max_{n_0\le t\le n_1}\hat\Delta_t$ using the original sequence $\{(x_1,y_1),...,(x_n,y_n)\}$, and let us denote its value as $S$. Then we use permutation/bootstrap to get a new sequence, say, $\{(x^1_1,y^1_1),...,(x^1_n,y^1_n)\}$, and we calculate $\max_{n_0\le t\le n_1}\hat\Delta_t$ on this new sequence. Say we get $S^{(1)}$. We repeat this re-sampling step for $m$ times and get $S^{(1)}, …, S^{(m)}$. The p-value is then calculated as the $1/m\sum_{i=1}^mI(S^{(i)} \ge S)$, i.e., the proportion of empirical samples that is larger than the observed value $S$. We note that this is also the approach used in many existing change point literature, for example, [1][2][3].
>
> **References**
>
> [1] Dubey, P. and Mu ̈ller, H.-G. Frechet change point detection. arXiv:1911.11864, 2019.
>
> [2] Chen, H., Zhang, N., et al. Graph-based change-point de- tection. The Annals of Statistics, 43(1):139–176, 2015
>
> [3] Chu, L., H. Chen, et al. (2019). Asymptotic distribution-free change-point detection for multivariate and non-euclidean data. The Annals of Statistics 47(1), 382–414.

---

### Official Review · Reviewer_pAEt · 2022-07-10

**Rating:** 6
**Confidence:** 3
**Soundness:** 3 good
**Presentation:** 3 good
**Contribution:** 3 good

**Summary:**

This paper addresses the problem of detecting abrupt changes in the conditional distribution of two (multivariate) random variables, using a finite time indexed data sample. The presented proposal entirely builds on a non-parametric approach. The presented approach is theoretically studied and some consistency results are provided. An empirical validation is also presented with both synthetic and real data sets.


**Questions:**

Could you please provide the reasons for the equation defining tilda delta at the beginning of page 6?

How realistic are the assumptions supporting your theoretical analysis?

Is Theorem 5.1 of relevance in the context of finite data sets?


**Limitations:**

Mostly yes. The only issue is the set of assumptions made for the theoretical results. See my question about.

**Strengths And Weaknesses:**

Strengths:
- The presented approach is quite general and its implementation is simple. It does not assume the existence of a likelihood model, in opposition to many previous proposals.

- Authors provide a non straightforward theoretical analysis of the consistency of the presented approach.

- The presented methodology applies to multidimensional problems, which is a problem not widely addressed in the existing literature.

Weaknesses:
- Some parts of the approach are not derived using well-defined principles. For example, at the beginning of Page 6: “we suggest setting” the tilde delta variable according to some equation whose rationale is not discussed and it is hard to understand.

- All the theoretical results established in this work are based on complex technical assumptions which are hard to interpret and, in many cases, impossible to verify.

- A central issue in this kind of problems is “how big” should be the “abrupt change” to be possible for the proposed method to detect it from a finite data sample. This is not explicitly addressed in this work.

- The computational complexity of the approach makes it impractical for many real-life applications.

---

> ### Author Response · Authors · 2022-08-02
> **Response to Reviewer pAEt**
>
> We thank the reviewer for the positive review and thoughtful questions!
>
> **Reasons for defining $\hat\Delta_t$ at the beginning of page 6**
> * We apologize for the confusion - we should have clarified the reasoning for the choice of $\hat\Delta_t$ and we will certainly do it in the final version. The motivation for defining $\hat\Delta_t$ is Theorem 5.1 in Section 5. In short, Theorem 5.1 studies the limiting behavior of $\tilde\Delta_t$, and says that its variance is bounded by the complicated multiplicative constant in the definition of $\hat\Delta_t$. So we use that constant to rescale $\tilde\Delta_t$.
> * The confusion might come from the way we organize the paper. Since the derivation of Theorem 5.1 requires a lot of technical assumptions, we deliberately keep it as a separate section in the end so readers can more easily focus and understand the main method. Since this is causing some confusion, we will make the cross-reference clearer in the final version.
>
> **How realistic are the assumptions**
> * Thank you for raising this important issue! Our assumptions are technical, but in general they are mild and not that restrictive. For example, when the space of $x$ is compact, and $k_X$ and $k_Y$ are Gaussian kernels, assumption 1,3,4 are all satisfied. If we further assume the space of $x$ is an Euclidean space with a density function, then assumption 5 is also satisfied. Assumption 2 is probably the hardest to give an intuition, but it essentially states that the conditional distribution $p(Y\mid X=x)$ changes smoothly in the sense that $p(Y\mid X=x_0)$ is close to $p(Y\mid X=x_1)$ when $x_0$ is close to $x_1$. We will add a paragraph that details these in the final version.
>
> **Relevance of Theorem 5.1 in the context of finite data sets**
> * The results in Theorem 5.1 are purely asymptotic; the theorem does not consider a finite sample size setup.
> * As the reviewer points out, a finite sample theoretical result would be needed if we were to tell ‘how big’ the abrupt change should be in order for the proposed method to detect it. However, in our nonparametric setup, the finite-sample theoretical analysis can be technically challenging. We agree with the reviewer  this issue is worth looking into, and a feasible trade-off is perhaps to derive a bound (either finite-sample or asymptotic) that explicitly depends on the signal-noise ratio (which will involve data distribution and kernel we choose in some way). This is an interesting area for future work.
>
> **Computational complexity**
> * We thank the reviewer for raising this important issue. The computational complexity of KCE is $O(n^2)$, and that for KCD is $O(n^3)$. In comparison, the computational complexity of our baselines ($D_Y, D_{XY}$, [1]) are all $O(n^2)$. The running time for both are acceptable in our simulations with a $n=1000$ sample size:  KCE takes about 1 second and KCD 30 seconds to calculate the corresponding statistics in R on a 2017 Macbook Pro. We agree that for larger datasets this can be slow, and developing faster algorithms is an important topic for subsequent work.
>
> **References**
>
> [1] Clive R Loader. Change point estimation using nonparametric regression. The Annals of 362 Statistics, 24(4):1667–1678, 1996.

---

### Official Review · Reviewer_ggSY · 2022-07-19

**Rating:** 6
**Confidence:** 4
**Soundness:** 3 good
**Presentation:** 3 good
**Contribution:** 3 good

**Summary:**

The paper studies a problem of detections and localizations of changes using the discrepancies between the Nadaraya-Watson estimator on data before and after a time. The paper develops the methods from scalar output cases to a general settings of outputs using kernel techniques.
To detect and localize the changes in data, a statistical test is proposed.


**Questions:**

- Let’s consider a case $t=1$, obviously ${\hat{f}}(t, \cdot)$ (\hat{f}{-}, sorry Latex does not work here) is a not-so-good estimator because it uses one data point. On the other hand $\hat{f}_{+}(t, \cdot)$ can be a reasonable estimation as it uses $n-1$ data points.  I think the statistical test should rely on presumably perfect estimators. It seems like $\tilde{\Delta}_t$ may not reflect well on the statistical test when there is a big difference in the number of left-side and right-side data. Of course, the variance $c_2(x)/[\rho (1-\rho)]$ can describe this. Do you have any results showing that the method can perform well when $\rho^*$ close $0$  or $1$ (i.e. $0.1$ or $0.9$)?

- As stated in Remark C.2, I wonder how to pick $c_2(x)$ in practice? It can greatly affect p-values.


**Ethics Review Area:**

["I don’t know"]

**Limitations:**




**Strengths And Weaknesses:**

Strengths:
- The paper is well-written in general. It states the problem clearly, giving solutions for simple tasks then the genel ones.
- The experiment results look interesting and promising. For example, in real world data, the method can detect change points which agree with actual events.

Weakness:
- Lack of discussion or emphasis on the transition from task I to task II. Specifically, it would be better to give an explanation as to why Kolmogorov’s entropy is involved. It might be technical but will provide a sense of directions for readers
- Missing some references of CP detection with Bayesian approaches. I believe the covariate shift would be related to this line of research, especially there is work using kernel maximum mean discrepancy to tackle the covariate shift problems that resembles the approach in this paper.

Reference:
Gretton et al. 2008. Covariate Shift by Kernel Mean Matching. In Dataset Shift in Machine Learning.

---

> ### Author Response · Authors · 2022-08-02
> **Response to Reviewer ggSY**
>
> We thank the reviewer for the positive review and thoughtful questions!
>
> **On more discussion on the transition from Task II to Task I**
> * Our general motivation for starting from Task II (the simplified problem) and later transitioning to Task I (the original problem) is that we find Task II to be a better understood problem in literature, and a problem that it is much easier to solve. The solution to Task II allowed us to immediately identify components (pairs of inner product between $y_i$ and $y_j$) which can be generalized to solve Task I by using an established kernel trick.
> * On why Kolmogorov’s entropy is involved: in short, it is an important concept used for the technical part of the paper, both for assumptions and proofs. We provide next more details. Recall that Kolmogorov’s entropy is needed when we standardize $\tilde\Delta_t$ so that it has the same asymptotic mean and variance across different $t$’s. To achieve that, we need to understand the limiting behavior of $\tilde\Delta_t$. Notice that for the simpler Task II, we can use Equation (6). For Task I, we have a similar expression
> $$\tilde\Delta_t=n^{-1}\sum_{i=1}^n||\hat f_-(t,x_i)-\hat f_+(t,x_i)||_{\mathcal{H}}^2 \quad\quad\quad (1)$$
>
> where $\hat f_-, \hat f_+$ are functions taking value in the infinite-dimensional space $\mathcal{H}$. Let us ignore all details here and focus on Equation (1). Understanding the limiting behavior of $\tilde\Delta_t$ ultimately requires deriving an upper bound. To bound each term in the sum in Equation (1) (say, the $i$-th term), we can derive a bound that depends on that particular $x_i$. But the $x_i$’s are random, and further we take the sum, so that bound will not work! In the end, we will need a ‘uniform bound’. To obtain such uniform bounds, regularity conditions must be placed on the space $x$’s. The Kolmogorov’s entropy is introduced to restrict the complexity of the space of $x$’s. We will add more discussion to this important point (thank you for raising it!) in the final version of the manuscript.
>
> **Missing reference of CP detection with Bayesian approaches**
> * Thank you for pointing out this lack of references! We will add them as a separate paragraph in Section 2 in the final version.
>
> **Performance of proposed statistics when change point is close to 0 or 1**
> * Thank you for raising this important issue. Yes, the performance of our method will be affected by the value of $\rho^*$. When it is close to 0 or 1, it does not perform as good when it is close to 0.5, for the reason pointed out by the reviewer. That is why we place a constraint for it to be bounded away from 0 and 1 in the sense that $0<\rho_0\le\rho^*\le\rho_1<1$, which is a common assumption in existing change point literature, see, for example, [1][2][3].
> * To better answer your question, we perform an additional experiment for KCE with $\rho^*=0.9$ and all other settings exactly the same as Experiment A in the paper (results with $\rho^*=0.7$ are directly taken from the paper):
> | $f_1(x)$ | $5x$ | $cos(x)$ | $x^2$ | $abs(x)$ | $0.1max(0,1-x)$ | $e^x$ | $1/[2(x+3)]$ |
> | ------ | ---- | ---- | ------| ----| ----------------| ---- | ---------- |
> | $\rho^*=0.7$ | $2.1\pm0.3$ | $2.7\pm 0.5$ | $4.9\pm 1.3$ | $2.9\pm 0.5$ | $6.4\pm1.6$ | $1.9\pm 0.4$ | $3.1\pm 0.6$ |
> | $\rho^*=0.9$ | $3.4\pm 0.9$ | $6.1\pm 1.9$ | $17.4\pm 6.3$ | $10.7\pm 2.6$ | $6.0\pm 2.0$ | $8.2\pm 2.7$ | $15.7\pm 6.8$ |
>
> Note that the performance does becomes worse from $\rho^*=0.7$ to $\rho^*=0.9$, but even for $\rho^*=0.9$ we find it to be good.
>
> **How to pick $c_2(x)$ in practice**
> * In practice we use permutations/bootstrap to calculate p-values, so $c_2(x)$ is not required. For purpose of standardization, we do not actually need $c_2(x)$ since it does not depend on $t$, and it always appears as a multiplicative constant for variance of $\Delta_t$’s, so it can be safely canceled out (in the same way we cancel out any constants).
>
> **References**
>
> [1] Dubey, P. and Mu ̈ller, H.-G. Frechet change point detection. arXiv:1911.11864, 2019.
>
> [2] Chen, H., Zhang, N., et al. Graph-based change-point de- tection. The Annals of Statistics, 43(1):139–176, 2015
>
> [3] Chu, L., H. Chen, et al. (2019). Asymptotic distribution-free change-point detection for multivariate and non-euclidean data. The Annals of Statistics 47(1), 382–414.

---

### Author Response · Authors · 2022-08-02
**Author Response to All Reviewers**

We thank all reviewers for their positive feedback and constructive comments!

We are happy to see that all reviewers agree that this work tackles a relevant problem to the NeurIPS community, provides a novel and general solution, with theoretical guarantees and promising empirical results.

We appreciate all comments pointing out insufficiencies and providing ideas for future improvements, including more explanations/examples for the technical assumptions, and generalizing to multiple or gradual change points. Below we answer each reviewer’s questions and issues raised as separate comments.

---

### Meta-Review · Area_Chair_izHR · 2022-08-24

**Recommendation:** Accept
**Confidence:** Certain

**Metareview:**

This manuscript enjoyed universal recommendation of acceptance from the reviewers after the initial review phase. The reviewers did note several minor issues in these initial reviews, many of which were resolved by insightful responses from the authors. I encourage the authors to edit the manuscript to reflect the insights gained from this interaction when preparing an updated version.

**Award:**

No

---

### Decision · Program_Chairs · 2022-09-14

Accept